# CNN and Deep Sets for End-to-End Whole Slide Image Representation Learning

**Sobhan Hemati**[1], **Shivam Kalra**[1], **Cameron Meaney** [2], **Morteza Babaie** [1], **Ali Ghodsi** [4,5], **H.R. Tizhoosh**[1,5]

[1] *Kimia Lab, University of Waterloo, Waterloo, ON, Canada*

[2] *Department of Applied Mathematics, University of Waterloo, Waterloo, ON, Canada*

[4] *Data Analytics Laboratory, University of Waterloo, Waterloo, ON, Canada*

[5] *Vector Institute, MaRS Centre, Toronto, ON, Canada*

**Editors:** Under Review for MIDL 2021

## Abstract

Digital pathology has enabled us to capture, store and analyze scanned biopsy samples as digital images. Recent advances in deep learning are contributing to computational pathology to improve diagnosis and treatment. However, considering challenges inherent to whole slide images (WSIs), it is not easy to employ deep learning in digital pathology. More importantly, computational bottlenecks induced by the gigapixel WSIs make it difficult to use deep learning for end-to-end image representation. To mitigate this challenge, many patch-based approaches have been proposed. Although patching WSIs enables us to use deep learning, we end up with a bag of patches or set representation which makes downstream tasks non-trivial. More importantly, considering set representation per WSI, it is not clear how one can obtain similarity between two WSIs (sets) for tasks like image search matching. To address this challenge, we propose a neural network based on Convolutions Neural Network (CNN) and Deep Sets to learn one permutation invariant vector representation per WSI in an end-to-end manner. Considering available labels at the WSI level - namely, primary site and cancer subtypes - we train the proposed network in a multi-label setting to encode both primary site and diagnosis. Having in mind that every primary site has its own specific cancer subtypes, we propose to use the predicted label for the primary site to recognize the cancer subtype. The proposed architecture is used for transfer learning of WSIs and validated two different tasks, i.e., search and classification. The results show that the proposed architecture can be used to obtain WSI representations that achieve better performance both in terms of retrieval performance and search time against *Yottixel*, a recently developed search engine for pathology images. Further, the model achieved competitive performance against the state-of-art in lung cancer classification.

**Keywords:** Whole-Slide Image Representation Learning, Whole-Slide Image Search, Multi-Instance Learning, Multi-label Classification, Digital Pathology

## 1. Introduction

The advent of digital pathology has provided researchers with a wealth of scanned biopsy samples. Accordingly, the amount of data stored in digital pathology archives has grown significantly as entire specimen slides or whole slide images (WSIs) can be imaged at once and stored as an digital images. The increased acquisition of this type of data has opened new avenues in the quantitative analysis of tissue histopathology, e.g., support the diagnostic

process by reducing the inter- and intra-observer variability among pathologists. Considering this, as well as other advantages of digital pathology (Niazi et al., 2019), one expects that histopathology images can be analyzed using the myriad computer vision algorithms currently available for similar tasks. As such, the usage of deep learning for WSI analysis has become an active area of research. Unfortunately, scientific progress with these data has been slowed because of difficulties with the data itself. These difficulties include highly complex textures of different tissue types, color variations caused by different stainings, rotationally invariant nature of WSIs, lack of labelled data and most notably, and the extremely large size of the images (often larger than 50,000 by 50,000 pixels). Additionally, these images are multi-resolution; each WSI may contain images from different zooming levels, primarily 5x, 10x, 20x, and 40x magnification (Tizhoosh and Pantanowitz, 2018).

The largest obstacle hindering the application of deep networks in computational pathology tasks is the sheer size of the images that makes it infeasible - or perhaps even impossible - to obtain a vector representation for a given WSI. In practice, this hurdle is often bypassed by simply considering small 'patches' of the WSI, a set of which is meant to represent the entire WSI (Faust et al., 2018; Chenni et al., 2019). Existing patching schemes allow us to split the WSI into tiles to be inputted to deep CNNs for WSI representation learning. However, such representations impose some new challenges. Firstly, significant memory resources are necessary to store sets of high dimensional vector representations for each WSI. Secondly, and more challenging, employing set representations for downstream problems, e.g., WSI classification and retrieval is not straightforward.

## 2. Related work

Considering gigapixel nature of WSIs, there is a large body of work on producing WSI representations suitable for different quantitative tasks. Authors in (Coudray et al., 2018) trained an Inception-V3 model on patches extracted from 20x and 5x magnifications for lung cancer subtype classification. To predict a label for a WSI from patch label predictions, they employed a simple heuristic based on the proportion of the patches assigned to each category. Hou et al. (Hou et al., 2016) proposed a patch-level classifier for WSI classification. In order to combine their patch-level predictions, they proposed a decision fusion model. By considering the spatial relationships between the patches, they utilized an expectation-maximization method to obtain the set of distinct patches from each WSI. Tellez et al. (Tellez et al., 2019) proposed a two-step method to employ CNNs for WSI classification. To this end, in the first stage they compress image patches using unsupervised learning. Then compressed patches are placed together (such that their spatial position is kept) and they are fed to another CNN for final prediction.

These and many other papers all used patch level training with decision fusion methods to achieve WSI level labels. Although this can be a helpful approach for classification, for many other tasks like search, it leads to a set of vector representations which have to be used to calculate distances between WSIs. There is no established way how to calculate distance between two sets of vectors. For example, authors in (Kalra et al., 2020b; Riasatian et al., 2021) resorted to the heuristic approach of taking the median of minimums to calculate the total distance between two WSIs. Although they were able to show that their approach achieved satisfactory performance, due to the computational complexity inherent to the

median of minimums method, the retrieval time can be considerably high unless binary encoding algorithms (Hemati et al., 2020) are used. On the other hand, representing a WSI using one vector not only removes the necessity of resorting to decision fusion methods in classification, but also considerably simplifies the WSI search problem.

In the context of WSI representation learning, different methods have been proposed to obtain one vector for representation of each WSI. For example, in Spatio-Net (Kong et al., 2017) patches are first processed by a CNN, then the embedded patches with each neighbor are fed into 2D-LSTM layers to capture the spatial information. Representing each WSI as bag of image patches makes multiple instance learning schemes (MIL) (Dietterich et al., 1997; Kalra et al., 2020a; Campanella et al., 2018) a natural approach to WSI representation (Quellec et al., 2017). More precisely, the mentioned patch-based CNN by Hou et al. (Hou et al., 2016) can be seen as a MIL method to determine instance classes. However, the two-stage neural networks and EM approach appeared to perform sub-optimally. Another recent work on MIL is attention -based MIL (Ilse et al., 2018) which was shown to be effective on medical data. Considering this, employing permutation invariant networks show potential as an effective approach for developing end-to-end WSI representation learning. One recent work on permutation invariant networks is Deep Sets (Zaheer et al., 2017). In the original work detailing Deep Sets, the authors specified a permutation-invariant function and proposed to employ universal set function approximators in neural network. They showed that despite its simplicity, their proposed permutation-invariant architecture can achieve promising performance in a variety of tasks including point cloud classification.

The objective of this paper is to propose an end-to-end permutation invariant CNN capable of obtaining a vector representation for a WSI. We use Deep Sets as a simple permutation-invariant neural network which makes it suitable for patch set data for WSI representation learning. We propose to employ a CNN along with Deep Sets to achieve a single global representation per WSI. To this end, we propose two reshape layers to connect our CNN to Deep Sets such that we can train a deep network in an end-to-end manner. Note that having one global representation for each WSI enables us to train our network in a multi-label classification scheme such that the targets for each WSI are primary site and primary diagnosis. This enables the proposed CNN-Deep Sets (CNN-DS) architecture to be used for WSI search in both horizontal search (search for primary site) and vertical search (search for primary diagnosis) (Kalra et al., 2020b). In order to further guide the proposed CNN-DS, we employ hierarchical multi-label training where primary site information is used to predict primary diagnosis labels. This idea is based on the fact that every primary site has its own disease sybtypes so we prevent the network from predicting meaningless diseases/primary site pairs. We show that the proposed network coupled with hierarchical multi-label training can be used for WSI representation. We validate the proposed scheme against Yottixel for the image search task on Cancer Genome Atlas (TCGA) dataset (Weinstein et al., 2013; Cooper et al., 2018) both in terms of retrieval performance and speed.

## 3. Material and Method

**Dataset.** We employ 5861, 281, and 604 WSIs unfrozen sections from TCGA for training, validation, and testing, respectively. The dataset spanned 24 primary sites and 30 primary

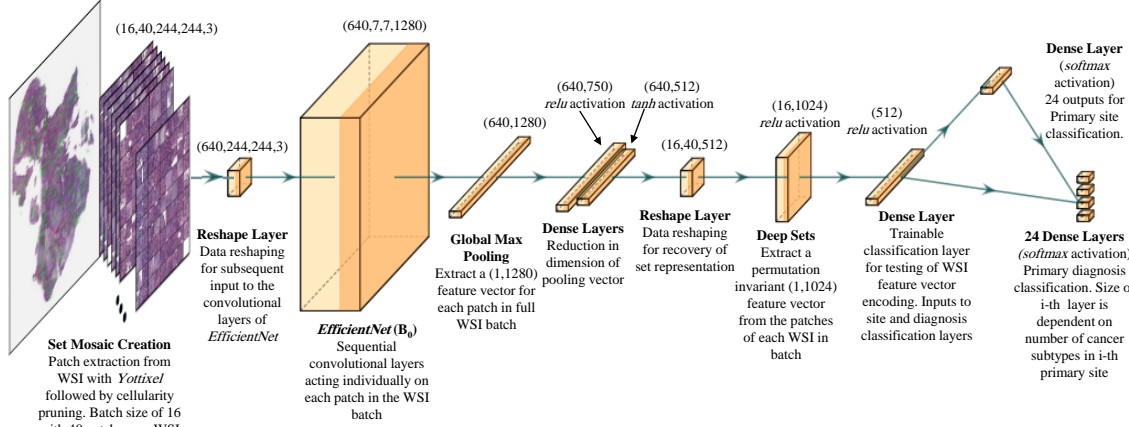

Figure 1: Proposed architecture for end-to-end WSI representation leaning.

cancer diagnoses. The tumour types available in the dataset include brain, breast, endocrine, gastrointestinal tract, gynecological, hematopoietic, liver/pancreaticobiliary, melanocytic, head and neck, prostate/testis, pulmonary, and urinary tract.

**Preprocessing.** The common practice to deal with gigapixel WSIs is patch extraction which leads to a bag of patches (set representation) (Tizhoosh and Pantanowitz, 2018). This type of image embedding pulls out some patches so that the network can train on a smaller set without sacrificing too much of tissue information. As it is not always clear which areas of a WSI are the regions of interest, patch extraction is challenging. In particular a chosen patch may not be relevant to the WSI diagnosis as it may contain exclusively healthy tissue or a combination of healthy and malignant tissue. Considering this, the patch extraction step is crucial as we may loose valuable information. This problem can be more severe in patch-based training as we assign WSI label to each patch which may not be correct. In this paper, we employ a patch extraction algorithm used in Yottixel (Kalra et al., 2020b). The patch selection method selects the representative patches from a WSI. We removed non-tissue portions of WSIs using colour threshold. The remaining tissue-containing patches are grouped into a pre-set number of categories through a clustering algorithm (we chose 9, and K-means algorithms). A portion of all clustered patches (e.g., 10%) are randomly selected within each cluster, yielding a mosaic. The mosaic is transformed into a set of features, obtained through a deep network (shown in Figure 1). The mosaic is meant to be representative of the full WSI, and enables much computational convenient computation for training of neural networks we randomly accept 40 patches from the mosaic.

## 4. The proposed architecture for CNN-Deep Sets (CNN-DS)

Deep Sets: (Zaheer et al., 2017) Representing a WSI by a mosaic of patches reduces our WSI to a set representation. Motivated by this, we propose applying Deep Sets to learn a permutation-invariant representation for each WSI in an end-to-end manner. The general architecture proposed in Deep Sets for representation of set $X$ that contains elements $x_1, x_2, \ldots x_n$ follows the following form, $f_X = \phi(\theta(x_1), \ldots, \theta(x_n))$, where, $f_X$ is the set rep-

resentation, $\theta$ is a non-linear mapping and $\phi$ is a pooling operation, including sum, mean, and max. In the Deep Sets paper, the authors proved that their proposed architecture was capable of acting invariantly and universally on set inputs approximate any set function (Zaheer et al., 2017). The universal invariance refers to the property that shuffling the input vector does not result in a change of the output vector; mathematically, for any reordering, $\pi(i)$: $F(\{x_1, x_2, x_3, \ldots, x_n\}) = F(\{x_{\pi(1)}, x_{\pi(2)}, x_{\pi(3)}, \ldots, x_{\pi(n)}\})$). In this paper we employ the max pooling operation for symmetric function part of Deep Sets as it it has been shown to be superior to other pooling layers for set representation learning (Zaheer et al., 2017).

**CNN-DS design for end to end training.** To have an end-to-end algorithm that learns high-quality permutation invariant representation per WSI, we employ EfficientNet B0 (Tan and Le, 2019) prior to the Deep Sets model. Figure 1 shows our proposed CNN-DS architecture.

Crucial to the design of our network are two reshape layers: one before the CNN (EfficientNet $B_0$ here) and one before the permutation-invariant Deep Sets. The first reshape layer is necessary to feed into the convolutional layers. Considering batch size of 16, extracting 40 patches per each WSI (set size= 40), and resizing patches from $1000 \times 1000$ to $224 \times 224$, the input tensor of our network has the shape (16,40,224,224,3). To feed this 5-dimensional tensor to the CNN-Deep Sets, we use the reshape layer to turn the input tensor into a 4-dimensional tensor with shape (640,224,224,3) so that in EfficientNet each patch is treated as an individual image and not part of a set. EfficientNet then transforms the data into shape (640,7,7,1280) which is further reduced to shape (640,1280) by the global max pooling. To prepare this matrix for Deep Sets, we process it with two dense layers. These layers reduce the dimensionality and apply a symmetric activation function $\tanh(\cdot)$ which is helpful before symmetric functions employed by Deep Sets. After these dense layers, the data shape is (640,512). To retain the set nature of the data, our second reshape layer changes the dimensionality to (16,40,512). This data is then given to Deep Sets to obtain a global representation for each WSI, which was represented by a set of patches. After Deep Sets we have a (16,1024) representation where each WSI has been embedded as a 1024 dimensional vector.

**CNN-DS design for multi-label training.** To update the network parameters, the vector embedding of the WSI outputted by Deep Sets is used in a multi-label classification task where labels are primary site and primary cancer subtypes. First, the output of Deep Sets is inputted to two different dense layers for primary site and cancer subtype classification. The elevated layer in Figure 1 is the primary site classifier component with 24 outputs and a softmax activation function where each output predict a primary site probability for the WSI. Since every primary site has its own cancer subtype, we can use the primary site predicted label to predict the primary diagnosis label. We therefore design the final lower layer to be a set of 24 layers associated with 24 primary sites where number of outputs for each layer is equal to the number of cancer subtypes for that primary site. For example, if the first layer in the lower final layer represents the brain as the primary site, then the primary diagnosis type layer will either be Glioblastoma Multiforme (GBM) or Lower Grade Glioma (LGG) - only two possible outputs with a softmax activation function. This layer therefore calculates the $P(\text{GBM}|\text{Brain})$ and $P(\text{LGG}|\text{Brain})$ probabilities. However, we aimed to calculate $P(\text{GBM})$ and $P(\text{LGG})$ probabilities with this

assumption that we know the probability of the given WSI is Brain, i.e., $P(\text{Brain})$ which we can obtained from upper final layer. To do this we use law of total probability as follows: $P(\text{GBM}) = P(\text{GBM}|\text{Brain})P(\text{Brain})$, and $P(\text{LGG}) = P(\text{LGG}|\text{Brain})P(\text{Brain})$ The multiplication between $P(\text{Brain})$ and $P(\text{GBM}|\text{Brain})$, or $P(\text{LGG}|\text{Brain})$ is shown using the connection between upper and lower final layers. We develop these layers for all other primary sites and their corresponding cancer subtypes where categorical cross entropy is used as loss function. Compared with a simple multi-label training with a sigmoid activation and binary cross entropy, this guided multi-label training needs significantly fewer epochs.

**Training.** All patches were reduced from 1000 by 1000 to 224 by 244 images. Then, for each WSI we ended up with a tensor of shape (40, 224, 244, 3) where 40 is number of patches per WSI. We set the batch size to 16 which leads to a tensor shape (16, 40, 224, 244, 3) for one batch of data. A batch of this size is quite large, leading to issues in regular GPU memory and run times. To handle data of this size we employed four Tesla V 100 GPUs in parallel mode. We employed the Adam optimizer (Kingma and Ba, 2014) with 0.000001 learning rate to avoid instabilities. The Albumentations library (Buslaev et al., 2020) was used to apply horizontal and vertical flip, 90 degree rotation, shifting and scaling data augmentation. Finally, in the last two dense layers we employed dropout at a 0.25 rate.

## 5. Results

To validate the proposed architecture for WSI representation, we employ the CNN-DS to obtain one feature vector for set of patches (here 40) per WSI. The output of the feature extractor for the proposed architecture is obtained from the dense layer after the Deep Sets layer, a 512 dimensional representation for each WSI. Unlike the training, obtaining WSI representations for test data can be done using a regular GPU. To investigate the quality of obtained WSI representations we validate the obtained features in the image search task for test data. We compare the proposed method with Yottixel search engine (Kalra et al., 2020b) on two different WSI search tasks, namely, horizontal and vertical search. Horizontal search refers to how accurate we can find the tumour type across the entire test database. Vertical search quantifies how accurately we find the correct cancer subtype of a tumour type among the slides of a specific primary site including different primary diagnoses. Due to small size of test set, we employ leave-one-out strategy and report the average scores.

**Search performance results.** The $k$-NN horizontal search results both for $k = 3$ and $k = 5$ are shown in Table 1. Clearly, almost in all primary sites there is a significant improvement in retrieval performance compared with Yottixel search engine. Table 2 presents the $k$-NN vertical search result using Yottixel and WSI embeddings obtained from CNN-DS. Unlike horizontal search, CNN-DS obtained better results in all cases compared with Yottixel in vertical search; in some cases Yottixel achieves better results. Looking more closely at these cases, the improvement of Yottixel against CNN-DS is not significant in most cases. Figure 2 shows the 2-D representation of obtained WSI embedding using CNN-DS labelled based on primary site and primary diagnosis labels.

**Lung cancer classification.** The Lung Adenocarcinoma (LUAD) and Lung Squamous Cell Carcinoma (LUSC) are two main cancer types of non-small cell lung cancer. The classification of LUAD versus LUSC can aid pathologists in diagnosis of these cancer

Table 1: Majority-3 and 5 search accuracy (%) for the horizontal search (primary site identification) among 604 WSIs for Yottixel and CNN Deep Sets (best results in green).

| Tumor Type | Patient # | Accuracy (in %) | | | |
| --- | --- | --- | --- | --- | --- |
| | | Yottixel ($k = 3$) | CNN-DS ($k = 3$) | Yottixel ($k = 5$) | CNN-DS ($k = 5$) |
| Brain | 46 | 73 | 91 | 73 | 89 |
| Breast | 77 | 45 | 77 | 38 | 79 |
| Endocrine | 71 | 61 | 66 | 59 | 62 |
| Gastro. | 69 | 50 | 75 | 49 | 74 |
| Gynaec. | 18 | 16 | 33 | 0 | 27 |
| Head/neck | 23 | 17 | 69 | 13 | 65 |
| Liver | 44 | 43 | 56 | 36 | 43 |
| Melanocytic | 18 | 16 | 50 | 5 | 38 |
| Mesenchymal | 12 | 8 | 100 | 0 | 83 |
| Prostate/testis | 44 | 47 | 81 | 43 | 77 |
| Pulmonary | 68 | 58 | 91 | 54 | 89 |
| Urinary tract | 112 | 67 | 76 | 62 | 74 |

Table 2: Majority-3 and -5 search through $k$-NN for the vertical search among 604 WSIs. Best F1-measure values highlighted.

| Site | Subtype | $n_{slides}$ | F1-measure (in %) | | | |
| --- | --- | --- | --- | --- | --- | --- |
| | | | Yottixel | CNN-DS | Yottixel | CNN-DS |
| Brain | LGG | 23 | 78 | 89 | 75 | 81 |
| | GBM | 23 | 82 | 89 | 83 | 84 |
| Endocrine | THCA | 50 | 92 | 98 | 91 | 98 |
| | ACC | 6 | 25 | 28 | 28 | 0 |
| | PCPG | 15 | 61 | 81 | 61 | 79 |
| Gastro. | ESCA | 10 | 12 | 44 | 25 | 55 |
| | COAD | 27 | 62 | 69 | 54 | 70 |
| | STAD | 22 | 61 | 64 | 57 | 78 |
| | READ | 10 | 30 | 55 | 16 | 0 |
| Gynaeco. | UCS | 3 | 75 | 80 | 50 | 50 |
| | CESC | 6 | 92 | 66 | 76 | 80 |
| | OV | 9 | 80 | 82 | 66 | 82 |
| Liver, panc. | CHOL | 4 | 26 | 0 | 25 | 0 |
| | LIHC | 32 | 82 | 95 | 87 | 95 |
| | PAAD | 8 | 94 | 94 | 77 | 94 |
| Prostate/testis | PRAD | 31 | 98 | 97 | 95 | 96 |
| | TGCT | 13 | 96 | 93 | 86 | 93 |
| Pulmonary | LUAD | 30 | 62 | 61 | 62 | 61 |
| | LUSC | 35 | 69 | 60 | 69 | 62 |
| | MESO | 3 | 0 | 50 | 0 | 0 |
| Urinary tract | BLCA | 31 | 89 | 95 | 86 | 94 |
| | KIRC | 47 | 91 | 87 | 89 | 84 |
| | KIRP | 25 | 75 | 84 | 79 | 81 |
| | KICH | 9 | 70 | 53 | 66 | 0 |

subtypes that include 65-70% of all lung cancers (Zappa and Mousa, 2016). To validate the performance of CNN-DS, we apply it to LUAD/LUAC classification task. We gathered 2,580 (H&E) stained WSIs of lung cancer from TCGA repository. Among this, we employ 1,806 for training set and the remaining 774 WSIs for test set (Kalra et al., 2020a). The patch selection and the architecture design of CNN-DS is the same as the one that used in transfer learning task. We avoid training convolutional layers to have a fair comparison

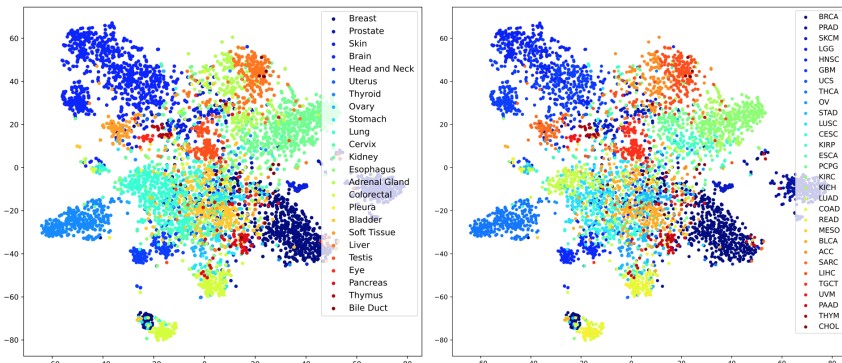

Figure 2: 2-D representation of obtained WSI embedding using CNN-DS labelled based on 24 primary sites (left) ad 30 primary diagnoses (right).

against other transfer-learning based methods. The results have been reported in 3 where CNN-DS can achieve competitive performance against the state-of-art.

Table 3: CNN-DS evaluation on lung cancer classification via transfer learning.

| Algorithm | Accuracy (in %) |
|---|---|
| Coudray et al. (Coudray et al., 2018) | 85 |
| Kalra & Adnan et al. (Kalra et al., 2020a) | 84 |
| Khosravi et al. (Khosravi et al., 2018) | 83 |
| Yu et al. (Yu et al., 2016) | 75 |
| CNN-DS (Ours) | 86 |

**Query time comparison against Yottixel.** We inteded to obtain one global representation for a WSI. We argued that this is particularly useful for WSI search as the set representation is bypassed. Hence, we measured query time for the leave-one-out approach used for 604 WSIs. Results showed that while for Yottixel it takes around 16 minutes to calculate pairwise distances between 604 WSIs, in our case it takes around 20 seconds to reproduce the results.

## 6. Conclusions

We employed Deep Sets along with a CNN for end-to-end WSI representation. This was inspired by bag of patches (set) representation per WSI. Two reshape layers connected CNN with Deep Sets. We propose to train our CNN-DS in the multi-label scheme. We used the law of total probability to capture the primary site predicted probability for obtaining probability of primary diagnosis. We validated the proposed topology in a transfer learning scheme for WSI search. We showed that the proposed architecture can obtain WSI embeddings leading to comparable retrieval performance compared with Yottixel while reducing the retrieval time significantly. We also applied the proposed scheme to lung classifcation task and achieved competitive results compared with the state-of-art.

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

In the following, the full description of the abbreviations for cancer subtypes in Table 2 have been presented in Table 4.

Table 4: Full description for primary diagnosis abbreviations used in the paper.

| Abbreviation | Primary Diagnosis |
|---|---|
| ACC | Adrenocortical Carcinoma |
| BLCA | Bladder Urothelial Carcinoma |
| CESC | Cervical Squamous Cell Carcinoma and Endocervical Adenoc. |
| CHOL | Cholangiocarcinoma |
| COAD | Colon Adenocarcinoma |
| ESCA | Esophageal Carcinoma |
| GBM | Glioblastoma Multiforme |
| KICH | Kidney Chromophobe |
| KIRC | Kidney Renal Clear Cell Carcinoma |
| KIRP | Kidney Renal Papillary Cell Carcinoma |
| LGG | Brain Lower Grade Glioma |
| LIHC | Liver Hepatocellular Carcinoma |
| LUAD | Lung Adenocarcinoma |
| LUSC | Lung Squamous Cell Carcinoma |
| MESO | Mesothelioma |
| OV | Ovarian Serous Cystadenocarcinoma |
| PAAD | Pancreatic Adenocarcinoma |
| PCPG | Pheochromocytoma and Paraganglioma |
| PRAD | Prostate Adenocarcinoma |
| READ | Rectum Adenocarcinoma |
| STAD | Stomach Adenocarcinoma |
| TGCT | Testicular Germ Cell Tumors |
| THCA | Thyroid Carcinoma |
| UCS | Uterine Carcinosarcoma |

