# OpenReview forum: "CNN and Deep Sets for End-to-End Whole Slide Image Representation Learning"
_MIDL.io/2021/Conference — MIDL 2021_

### Official Review · AnonReviewer1 · 2021-03-07

**Confidence:** 4
**Preliminary Rating:** 2
**Final Rating:** 2

**Summary:**

The paper addresses the problem of whole slide image (WSI) representation learning. In particular, it employs a CNN along with Deep Sets, a simple permutation-invariant neural network, to achieve a single global representation for WSI. The method is evaluated on Cancer Genome Atlas (TCGA) dataset and compared with Yottixel’s method in terms of retrieval performance and speed.

**Strengths:**

+ The proposed framework is technically sound and easy to implement.
+ It is an interesting topic, which is very challenging and notoriously difficult to address.
+ This paper is clearly written and easy to follow.


**Weaknesses:**

- The proposed method is only compared with one method.
- The proposed method is only evaluated on one dataset.
- Some methods which are related to the proposed method are cited but not compared, e.g., Hemati’s work [1].

[1]. Sobhan Hemati, Mohammad Hadi Mehdizavareh, Shojaeddin Chenouri, and Hamid R
Tizhoosh. A non-alternating graph hashing algorithm for large-scale image search. arXiv
preprint arXiv:2012.13138, 2020.


**Deanonymize Review:**

no

**Detailed Comments:**

1. There is a missing citation in Related Work, 'For example, authors in (Kalra et al., 2020b; ?) resorted to the…’

2. Why is the learning rate set so small, 0.000001? Have you tried other settings?

3. “Due to small size of test set, we employ leave-one-out strategy and report the average scores as final results.” However, there are 5861, 281, and 604 WSIs unfrozen sections from TCGA for training, validation, and testing, respectively. Why still use the leave-one-out strategy?


**Final Rating Justification:**

The reviewer appreciates the effort the authors put into the revision. The authors add another experiment on lung cancer classification. However, there is no significant improvement compared to Coudray's method. And there is no discussion and analysis about this experiment. Also, I think the technical novelty in this paper is very limited. Thus, the reviewer keeps the rating unchanged.

**Justification Of The Preliminary Rating:**

The proposed approach for end-to-end training of WSI representation is interesting. However, this framework is built on existing build blocks, i.e., CNN and Deep Set.  So, the proposed method is limited regarding the technical novelty. More comparisons with other methods are also needed to show the advantages and effectiveness of the proposed method.

**Paper Type:**

validation/application paper

**Special Issue:**

no

---

> ### Author Response · Authors · 2021-03-17
> **Reply to AnonReviewer1**
>
> Thank you for your time and valuable comments. We are glad you found the paper to be technically sound, clearly written, and easy to follow. In preparing the final submission, we have added all the necessary changes as per your review. The changes have been highlighted in the manuscript.
>
> ### Addressing Weaknesses
>
>
> 1. *Q: The proposed method is only compared with one method. ... The proposed method is only evaluated on one dataset.* \
> Following your comment, we evaluated our method, CNN-DS on the lung classification task comparing it with 4 other methods in the literature. The results are presented in **Table 3, Pg 8**. The results show that the proposed method achieves highly competitive classification performance.
>
>
> 2. *Q: Some methods which are related to the proposed method are cited but not compared, e.g., Hemati’s work [1].* \
> We did not compare it with the cited paper because it does not provide a complete framework for WSI search. However, we cited the method as it can be used to further improve the WSI search speed of the proposed method which could be the subject of future studies.
>
>
> 3. There are two main contributions of the paper **(Pg. 5, two headlines have been added)**:
>      - We propose an end-to-end scheme for learning a single vector representation of a WSI using its mosaic. The single vector representation of a WSI significantly improves the search speed without affecting the search accuracy.
>     - We use a novel multi-label training while incorporating the hierarchical relationship between the primary site and the primary diagnosis of a WSI. The hierarchy of the labels is modeled as a two-branch output from the network, where the primary diagnosis probability is conditioned on the primary site probability. This architecture enables us to remove the redundancies of predicting unrelated primary diagnoses for a given primary site, therefore resulting in a more effective representation of a WSI.
>
>
> ### Addressing Detailed Comments
> 1. *Q: There is a missing citation in Related Work* \
> Thanks for reporting this. We added the citation in **Section 2, Pg. 2.**
>
> 2. *Q: Why is the learning rate set so small, 0.000001? ...* \
> To avoid instabilities during the training, we chose a small learning rate of 0.000001 and trained for more epochs (500 in our case). Added details in **Section 4, Pg. 6, Subsection: Training.**
>
> 3. *Q: ...Why still use the leave-one-out strategy?* \
> We performed the leave-one-out search on the test images to validate the efficacy of the proposed architecture in the transfer learning scheme. In other words, we are only validating our method using the WSIs of patients that have never been shown to the model (patients of the train and test dataset are disjoint sets).

---

### Official Review · AnonReviewer4 · 2021-03-08

**Confidence:** 4
**Preliminary Rating:** 3
**Recommendation:** Oral, Poster

**Summary:**

In the paper "CNN and Deep Sets for End-to-End Whole Slide Image Representation Learning " the authors propose a way to create WSI representation by CNN. The proposed approach is applied to searching WSI. The possibility to find similar slides among  WSI is a very welcome tool, that can find many practice applications. The authors compared the proposed solution with the state-of-the-art method, shows significant improvement.

**Strengths:**

The authors focused on solving an important task in computational pathology, that is fast WSI searching. The performed experiments show method potential and possibility. Moreover, the authors compared their approach with another available method shows significant improvement.

**Weaknesses:**

The method description should be extended. It is not clear why mosaic is created from 40 patches? How method will works if we use 64/32/16/8 patches? Will be faster?

Lack of details about applied data augmentation.

The "related work" section does not include sufficient information.  A few very important approaches, such as [1] or [2] are missed.

[1]Tellez D, Litjens G, van der Laak J, Ciompi F. Neural Image Compression for Gigapixel Histopathology Image Analysis. IEEE Trans Pattern Anal Mach Intell. 2021 Feb;43(2):567-578. doi: 10.1109/TPAMI.2019.2936841. Epub 2021 Jan 8. PMID: 31442971.

[2]Campanella, Gabriele, Vitor Werneck Krauss Silva, and Thomas J. Fuchs. "Terabyte-scale deep multiple instance learning for classification and localization in pathology." arXiv preprint arXiv:1805.06983 (2018).



**Deanonymize Review:**

no

**Detailed Comments:**

1. WSI presents not only biopsies
2. captions on figures should be readable. now they are too small.

**Justification Of The Preliminary Rating:**

The authors focused on solving an important task in computational pathology, that is fast WSI searching. That topic can be very interesting for the research community.  The method achieved very good results and was compared with another available solution.

**Paper Type:**

methodological development

**Special Issue:**

yes

---

> ### Author Response · Authors · 2021-03-17
> **Reply to AnonReviewer4**
>
> Thank you for your time and valuable comments.  We are glad you found the problem addressed in the paper and the idea to be interesting. In preparing the final submission, we have added all the necessary changes as per your review. The changes have been highlighted in the manuscript.
>
> 1. *Q: The method description should be extended.* \
> We added more details about the patch selection method in **Section 3 (Pg. 4, Subsection: Preprocessing).** The formulation of Deep Sets is also added in **Section 4 (Pg. 4 and 5).**
>
> 2. *Q: ...It is not clear why mosaic is created from 40 patches? How method will works if we use 64/32/16/8?...* \
> We chose a mosaic size to be 40  due to the limitation of GPU memory. Even the modern GPUs have around 32 GB VRAM. Selecting more than 40 patches from WSIs will have higher chances of covering more cancerous regions. We accept the limitation that 40 patches may not be enough to capture all the necessary information from a WSI. We plan to improve this limitation in future studies. We added a discussion about this weakness in  **Section 6 (Conclusions, Pg. 8).**
>
> 3. *Q: Lack of details about applied data augmentation.*
> We provided more information about data augmentation steps in the manuscript in **Section 4 (Pg. 6, the last 3 lines of the second paragraph).**
>
> 4. Thank you for letting us know about the interesting related works. We added and discussed them in **Section 2, Pg. 2 and 3.**
>
> **Note:** We added one additional experiment to compare our method against the existing state-of-the-art methods for the lung cancer sub-type classification task. The results are presented in **Table 3, Pg. 8.** The results show that the proposed method achieves highly competitive performance.

---

### Official Review · AnonReviewer3 · 2021-03-08

**Confidence:** 4
**Preliminary Rating:** 2
**Final Rating:** 3

**Summary:**

This paper presents a method to obtain a single vector from a WSI, which is descriptive enough to perform for example search of similar WSIs.
The method is partly based on Deep Sets method on permutation-invariant neural networks.
Patch extraction based on previous work (Yottixel method), arranging patches into a 'mosaic' of 40 patches using clustered patches.
Validated using an existing search engine named Yottixel, based on TCGA.
Authors show substantial speed-up in a search problem compared to previous work (e.g., Yottixel).

**Strengths:**

* Modelling the slide as one vector improves search speed over existing work based on bags of patches to represent whole-slide images
* The proposed approach outperforms previous work (e.g., Yottixel) in searching similar cases in most primary sites of TCGA.

**Weaknesses:**

* One reference is missing (?) in section 2
* I don't understand what is the advantage of making a mosaic with 40 images and make batches with 16 of those if in the end all patches are treated independently by EfficientNet as 640 different samples
* The Yottixel method is heavily used in this paper, from patch sampling and clustering to using its search functionalities. However, this makes the paper lack many technical details (for example, patch sampling and clustering techniques) and also makes it difficult to clearly appreciate what the novel contribution of this work is since it heavily relies on previous work. If the novelty is in the use of a single vector to represent the slide instead of a bag of patches, then this paper would have benefitted from a more detailed comparison between the two strategies, focusing on failures of one or the other method, for example. Why are there 46 brain patients and CNN-DS finds 91? Are the 45 additional cases some false positives? A confusion matrix would have helped, as well as some discussion on how to further improve the presented method.
* The Deep Sets method is used, based on previous work, but is not explained in this paper
* The use of a permutation invariant approach based on Deep Sets is not fully justified, and it is not clear why other approaches like pooling would not work here
* No visual examples are shown, no discussion on cases of failure and possible reasons why this happens


**Deanonymize Review:**

no

**Final Rating Justification:**

I would like to thank the authors for addressing my comments.

**Justification Of The Preliminary Rating:**

This paper presents a method to compute a single vector from a whole-slide image, to be used to find similar images on TCGA. The method show superiority when compared to other published work, but it also relies on previous work and does not fully explain the reasons for improved results, as well as a detailed analysis on achieved results and possible future improvement.

**Paper Type:**

validation/application paper

**Special Issue:**

no

---

> ### Author Response · Authors · 2021-03-17
> **Reply to AnonReviewer3**
>
> Thank you for your time and valuable comments. In preparing the final submission, we have added all the necessary changes as per your review. The changes have been highlighted in the manuscript.
>
> 1. Thanks for reporting this typo. We fixed it in the updated manuscript **(Pg. 2, last para.)**.
>
> 2. *Q:  ...Why are there 46 brain patients and CNN-DS finds 91? Are the 45 additional cases some false positives?...* \
> We clarified the results presented in Tables 1 and 2 in the caption **(Page 7, Tables 1 and 2 )**. For the 46 brain patients, CNN-DS (our approach) achieves 91% accuracy whereas Yottixel achieves 73%. The quantities, 91%, and 73% are the accuracy values not the number of retrieved WSIs.
>
> 3. *Q: I don't understand what is the advantage of making a mosaic with 40 images and make batches with 16  ...* \
> Here 16 represents the number of WSIs, and 40 represents the number of patches in each WSI as shown in **Figure 1, Pg. 4**. Firstly, the patches are processed individually by EfficientNet to extract feature vectors. Then these feature vectors are rearranged as a set, one set for each WSI. Finally, this set of feature vectors is processed in its entirety by the Deep Sets. This entire pipeline is trained end-to-end.
>
> 4. *Q: The Yottixel method is heavily used in this paper, from patch sampling and clustering to using its search functionalities. However, this makes the paper lack many technical details...* \
> We added more details about the patch selection method in **Section 3 (Pg. 4, Subsection: Preprocessing)**.  The formulation of Deep Sets is also added in **Section 4 (Pg. 4 and 5)**.
>
> 5. *Q: ...it difficult to clearly appreciate what the novel contribution of this work is since it heavily relies on previous work...*\
> There are two main contributions of the paper **(Pg. 5, two headlines have been added)**:
>      - We propose an end-to-end scheme for learning a single vector representation of a WSI using its mosaic. The single vector representation of a WSI significantly improves the search speed without affecting the search accuracy.
>     - We use a novel multi-label training while incorporating the hierarchical relationship between the primary site and the primary diagnosis of a WSI. The hierarchy of the labels is modeled as a two-branch output from the network, where the primary diagnosis probability is conditioned on the primary site probability. This architecture enables us to remove the redundancies of predicting unrelated primary diagnoses for a given primary site, therefore resulting in a more effective representation of a WSI.
>
> 6. *Q: The use of a permutation invariant approach based on Deep Sets is not fully justified, and it is not clear why other approaches like pooling would not work here* \
> The Deep Sets paper mathematically proves that a composition of non-linear mapping followed by symmetric function is a universal approximation of any set function. We directly utilized this proof for learning the WSI representation from its mosaics.  Other approaches would also work if they can be implemented in form of the equation in the Deep Sets paper. The formulation of Deep Sets is added in **Section 4 (Pg. 4 and 5)**.
>
> 7. *Q: ...no discussion on cases of failure and possible reasons why this happens...* \
> We added more details regarding the cases of failure and the possible reasoning. One of the limitations is the small number of patches may not capture the intrinsic variability for some of the WSIs. Changes are added in **Section 6 (Conclusions, Pg. 8)**.
>
> **Note:**  We added one additional experiment to compare our method against the existing state-of-the-art methods for the lung cancer sub-type classification task. The results are presented in **Table 3, Pg. 8**. The results show that the proposed method achieves highly competitive performance.

---

### Official Review · AnonReviewer2 · 2021-03-10

**Confidence:** 4
**Preliminary Rating:** 2

**Summary:**

This paper proposes a method to learn an invariant representation of a Whole Slide Image (WSI) using a set of representative patches. The patches are extracted from the WSI, and then used to build an invariant representation via convolutional neural networks (cnn) and deep sets. The resulting representation is validated in a image search task. The authors report competitive searching performance results when comparing with state-of-the-art automated tools for image search.

**Strengths:**

Interesting idea applying the representation invariance capability of deep sets to mosaics of patches extracted from a Whole Slide Image. The method was compared with a state-of-the-art search engine with competitive results. The data split also is properly described in the manuscript.

**Weaknesses:**

Several elements of the method are not described in detail, and it would allow for increasing the reproducibility of the presented methodology:
- The initial selection of the image via clustering is not properly described
- The formulation of deep sets is missing in the manuscript

**Deanonymize Review:**

no

**Detailed Comments:**

Several experiments would help testing the hypothesis that the representation is properly "set invariant":

1) The initial selection of the image cluster that is not described in detail. Is this clustering process deterministic? If it is not deterministic how variable is the invariant representation for the selection of different mosaics representing a particular WSI?

2)A particular experiment to test invariance would be to randomly shuffle the mosaic for several WSI, and report some distance metric across the generated vector

**Justification Of The Preliminary Rating:**

The idea of using invariant representation provided by deep sets in WSI is interesting. However, I think that additional experiments evaluating the invariance of the representation is missing in the paper

**Paper Type:**

methodological development

**Questions To Address In The Rebuttal:**

- Where is the evidence to support the representation invariance of the embedding vector?  At this stage, it is assumed that such representation is invariant because deep sets are used on the deep learning architecture. However, such hypothesis is not actually validated in the experimental setup

- It is not clear the impact of the initial clustering process for patch selection on the resulting embedding.



**Special Issue:**

no

---

> ### Author Response · Authors · 2021-03-17
> **Reply to AnonReviewer2**
>
> Thank you for your time and valuable comments. We are glad you found our idea to be interesting. In preparing the final submission, we have added all the necessary changes as per your review. The changes have been highlighted in the manuscript.
>
> 1. *Q: Where is the evidence to support the representation invariance of the embedding vector? ...* \
> The empirical evidence to support the permutation invariance of our model is not required.  The model is mathematically invariant to the permutation of its input, therefore experiments are not required to be designed to samples from any specific orderings as any change in the ordering of the input patches results in the same representation (the embedding vector). We clarified the above permutation-invariance property in the revised paper in **Section 4 (Pg. 4 and 5).**
>
> 2. *Q: It is not clear the impact of the initial clustering process for patch selection...* \
> (Kalra et al., 2020a) have established that there is not a significant change in the search performance of Yottixel even though the patch selection is stochastic in nature. Based on their findings we expect the embedding to be equally effective for any given run of the patch selection on the same WSI.
>
> 3. *Q: The initial selection of the image via clustering is not properly described* \
> We added more details about the patch selection method in **Section 3 (Pg. 4, Subsection: Preprocessing)**.
>
> 4. *Q:  The formulation of deep sets is missing in the manuscript* \
> The formulation of Deep Sets has been added in **Section 4 (Pg. 4 and 5)**.
>
> **Note:** We added one additional experiment to compare our method against the existing state-of-the-art methods for the lung cancer sub-type classification task. The results are presented in **Table 3, Pg. 8**. The results show that the proposed method achieves highly competitive performance.

---

### Meta-Review · Area_Chair1 · 2021-03-24

**Recommendation:** Accept (Poster)

**Metareview:**

Initially, the majority of reviewers suggested a weak rejection. After the rebuttal, one reviewer changed their opinion to weak accept, giving an even split between weak reject and weak accept. I do think the authors did an adequate rebuttal, specifically the new experiment on lung cancer to compare against the state-of-the-art. As such I propose acceptance as a poster presentation. I do agree with the comment by Reviewer 1 that some additional discussion on the final experiment should be added to the camera-ready paper.

**Paper Type:**

methodological development

---

### Decision · Program_Chairs · 2021-03-31

Accept